# Array of Miniaturized Amperometric Gas Sensors Using Atomic Gold Decorated Pt/PANI Electrodes in Room Temperature Ionic Liquid Films

**DOI:** 10.3390/s23084132

**Published:** 2023-04-20

**Authors:** Anifatul Faricha, Shohei Yoshida, Parthojit Chakraborty, Keisuke Okamoto, Tso-Fu Mark Chang, Masato Sone, Takamichi Nakamoto

**Affiliations:** 1Department of Information and Communications Engineering, Tokyo Institute of Technology, Yokohama 226-8503, Kanagawa, Japan; 2Department of Materials Science and Engineering, Tokyo Institute of Technology, Yokohama 226-8503, Kanagawa, Japan; 3Institute of Innovative Research, Tokyo Institute of Technology, Yokohama 226-8503, Kanagawa, Japan

**Keywords:** atomic gold decorated Pt/PANI electrodes, butanol isomers gas measurement, catalytic activity, miniaturized electrochemical sensors, room temperature ionic liquid

## Abstract

Miniaturized sensors possess many advantages, such as rapid response, easy chip integration, a possible lower concentration of target compound detection, etc. However, a major issue reported is a low signal response. In this study, a catalyst, the atomic gold clusters of Au_n_ where n = 2, was decorated at a platinum/polyaniline (Pt/PANI) working electrode to enhance the sensitivity of butanol isomers gas measurement. Isomer quantification is challenging because this compound has the same chemical formula and molar mass. Furthermore, to create a tiny sensor, a microliter of room-temperature ionic liquid was used as an electrolyte. The combination of the Au_2_ clusters decorated Pt/PANI and room temperature ionic liquid with several fixed electrochemical potentials was explored to obtain a high solubility of each analyte. According to the results, the presence of Au_2_ clusters increased the current density due to electrocatalytic activity compared to the electrode without Au_2_ clusters. In addition, the Au_2_ clusters on the modified electrode had a more linear concentration dependency trend than the modified electrode without atomic gold clusters. Finally, the separation among butanol isomers was enhanced using different combination of room-temperature ionic liquids and fixed potentials.

## 1. Introduction

The demand for gas sensors shows positive growth and is projected to increase at a compound annual growth rate (CAGR) of 8.9% from 2022 to 2030 in the global gas sensor market [1,2]. Several factors are responsible for the rise in demand for gas sensors, i.e., the development of miniaturized sensors, the improvement of technology, such as the internet of things (IoT), cloud computing, etc. [1,2,3]. Among types of sensors, the electrochemical (EC) sensor has been favorable due to high selectivity and less power consumption [1,2,3,4,5]. Numerous works have been conducted to improve EC sensors in terms of selectivity, sensitivity, and response time; the current major sensor design deposits all the electrodes (i.e., working electrode (WE), counter electrode (CE), and reference electrode (RE)) onto a small single substrate [6].

Furthermore, to increase the signal-to-noise ratio of the miniaturized sensor, the WE has been frequently redesigned; from a macro disk electrode (MDE) to a microelectrode array (MEA), and the latest is an interdigitated array (IDA) electrode [7,8]. In addition, it is impossible to use the aqueous electrolyte as a thin film because of its fluidity and volatility. Room temperature ionic liquid (RTIL) is a promising substitute for the conventional aqueous electrolyte; its high viscosity removes the evaporation and fluidity issues [5,7]. Moreover, only a small amount of RTIL can be used to realize a thin and small sensor [9]. However, the diffusion transport issue leads to a slow response time and decreased sensitivity due to its inherent physical high viscosity property [10,11].

Recently, it has been extensively reported that atomic gold clusters (size of cluster less than 2 nm referred to as an atomic cluster) have gained much attention because their physical and chemical properties dramatically differ from those of gold nanoparticles and bulk gold; for example, the atomic gold clusters not only have catalytic activity but also oscillation property, known as an odd-even pattern due to electron pairing effect [12,13,14,15,16]. Even-numbers are more stable than odd-number of atomic gold clusters; atomic clusters built from 2 and 6 gold atoms have the lowest second-order difference in total energy and the largest HOMO-LUMO energy gap and dissociation energy, which result in their high stability [15,16,17,18]. Furthermore, it is well known that the support matrix also plays a vital role in doping the atomic gold. In this study, polyaniline (PANI) was selected as the support matrix. PANI has been one of the most extensively studied conducting polymers; the literature sources regarding this polymer in acid or base, even in RTIL, are available [19,20,21,22,23]. In addition, PANI can be easily prepared and highly stable; moreover, it also has rich information and complex redox reaction. PANI exists in three redox states, i.e., leucoemeraldine (LB), emeraldine (EB), and pernigraniline (PB) [24,25]. It was confirmed that both PANI-modified electrodes and PANI doped with metal showed electrocatalytic activity [15,19]. Besides the support matrix, the catalytic performances of atomic gold clusters also depend on the atomic gold deposition process. In this study, we used potassium tetrachloroaurate (KAuCl_4_) as the gold precursor to supply the noble metal Au. Furthermore, to rinse away the excess ${\mathrm{AuCl}}_{4}^{-}$, perchloric acid (HClO_4_) was selected because it can minimize the PANI degradation during the gold deposition process compared to HCl [26].

Biofuels have become alternative energy sources to fulfill energy demand and support green energy [27,28]. Several biofuel productions are synthesized from biomass like bio-ethanol, bio-butanol, bio-diesel, etc. However, bio-butanol is preferable because it has a higher energy content, is non-corrosive in nature, releases lesser quantities of carbon monoxide upon combustion, etc. [27,29]. Four isomers exist, but only three are promising candidates for bio-butanol production, i.e., 1-butanol, isobutanol, and 2-butanol [30]. Gas measurements using isomer target compounds are challenging because they have exactly the same molecular formula and molar mass [31]. 

In our previous research, in a bulky EC system, it was successfully demonstrated and verified that modifying platinum (Pt) WE with PANI and atomic gold clusters (Au_n_ where n = 1 to 4) showed the electrocatalytic activity and exhibited an odd–even pattern for isomer compounds, particularly for the Au_2_ clusters [32,33,34]. Afterward, a tiny planar IDA electrode modified with PANI decorated Au_2_ clusters was successfully conducted and first reported by our research group; however, it used only one RTIL (1-ethyl-3-methylimidazolium trifluoromethanesulfonate) and one fixed EC potential (i.e., +1 V) [35,36]. In this study, we developed the miniaturized EC sensor array using the Au_2_ clusters decorated Pt/PANI in three RTILs with different physical-chemical properties. In addition, the modified Pt/PANI/Au_2_ on IDA electrodes, along with applying several fixed EC potentials, were expected to enhance the sensitivity of butanol isomers gas measurement and provide a higher electrocatalytic activity. Then, the single sensor was extended to the sensor array using several RTILs and EC potentials.

## 2. Materials and Methods

### 2.1. Chemical Compounds

This study used butanol isomers, i.e., 1-butanol, isobutanol, and 2-butanol, as the target compounds. All butanol isomers purchased from Sigma Aldrich Ltd., Tokyo, have a 74 g/mol molar mass and chemical formula of C_4_H_10_O. Furthermore, imidazolium, the common RTIL, was used as the electrolyte without further purification (purchased from Tokyo Chemical Industry Ltd., Tokyo, Japan). Three different RTILs having the same cations and different anions were selected, i.e., 1-ethyl-3-methylimidazolium acetate, 1-ethyl-3-methylimidazolium trifluoromethanesulfonate, 1-ethyl-3-methylimidazolium chloride. Table 1 provides detailed information regarding the physical and chemical properties of three RTILs. 

### 2.2. Sensor Fabrication

#### 2.2.1. Polymerization of Polyaniline (PANI)

The commercial IDA electrode used in this experiment is depicted in Figure 1a and purchased from BAS Co., Ltd., Tokyo, Japan. Afterward, PANI was polymerized on WE, shown in Figure 1b. The electropolymerization of PANI used 0.1 M aniline (C_6_H_5_NH_2_) in 2 M tetrafluorobic acid (HBF_4_) and the Galvanostatic method with a constant current of 0.56 mA for 260 s. The scanning electron microscope (SEM) image of PANI dendrite-like growth on the Pt planar IDA electrode is available in Figure 1c.

#### 2.2.2. Atomic Gold Deposition

Figure 2a depicts a brief diagram of the gold deposition system. The gold precursor for supplying noble metal Au was 0.2 mM KAuCl_4_, and 0.1 M HClO_4_ solution was used to rinse away the excess gold anions. The gold precursor and buffer solution flow were controlled through the solenoid valve. Afterward, the modified Pt/PANI electrode was installed in the chamber where its two WEs and CE were connected to the line of WE and CE at potentiostat, respectively (the potentiostat was based on IC TL074, Texas Instruments, Dallas, TX, USA); since the RE from IDA made of Pt, hence, external Ag/AgCl was used as RE to maintain the stable EC reaction during the gold deposition process which connected to the RE line at potentiostat. Furthermore, the timing diagram of the deposition process to fabricate the Au_2_ clusters is shown in Figure 2b [32,33,34,35], and modified Pt/PANI/Au_2_ using IDA electrode is described in Figure 2c.

Since the current commercial SEM technology is not able to be used to confirm the presence of atomic gold in PANI due to the resolution limits, the alternative way to confirm the Au_2_ clusters, which have also been massively reported and applied, has been using the information of oxidation from propanol isomers in the alkaline medium [15,19,26,32,33,34]. Figure 3 depicts the cyclic voltammogram (CV) curve of 1-propanol (0.5 M) and 2-propanol (0.5 M) in 1 M KOH generated from Pt/PANI/Au_2_, Pt/PANI/Au_1_, and Pt/PANI/Au_0_ (without atomic gold clusters); the scan rate was 50 mV/s, 10 CVs were performed, and only the last CV showed. 

As shown in Figure 3, Pt/PANI/Au_2_ exhibited catalytic activity both for 1-propanol and 2-propanol, agreeing with our previous result in the bulky EC system and with atomic gold obtained by other researchers [15,19,26,32,33,34]. Figure 3 shows that Pt/PANI/Au_2_ had the highest current density. For 1-propanol, Pt/PANI/Au_2_ exhibited two oxidation peaks at the forward scan at −0.1 V and +0.25 V, while only one oxidation peak at −0.1 V was obtained using Pt/PANI/Au_1_ and Pt/PANI/Au_0_. For 2-propanol, two oxidation peaks occurred at −0.1 V and +0.25 V using Pt/PANI/Au_2,_ whereas no oxidation peak was achieved in the forward scan using Pt/PANI/Au_0_ and only one oxidation peak at −0.1 V was achieved by Pt/PANI/Au_1_. Furthermore, using Pt/PANI/Au_2_, 1-propanol had a higher current density at the 1st oxidation peak than the 2nd oxidation peak, while 2-propanol had a higher current density at the 2nd peak than the 1st peak. Thus, we confirmed the deposition of Au_2_ clusters on Pt/PANI; to check the reproducibility of Au_2_ clusters decorated Pt/PANI, 3 different IDA electrodes were applied, and the results were available in Appendix A.

### 2.3. Measurement System

Figure 4 depicts the experimental setup used in this study. The carrier gas was nitrogen (N_2_), and the mass flow controller was set to 200 mL/min to control the N_2_ flows. The odor delivery system (ODS) controlled by the field programmable gate array (FPGA) board had eight channels where butanol isomers were located (each 6 mL of analyte was put at a vial in the ODS); detailed information related to ODS is available in [46]. The gas flow was recorded by two sensor types, i.e., quartz crystal microbalance (QCM) and IDA. QCM is a microgravimetric sensor; it is a mass-sensitive device, has high stability, and generally gives a linear response to the concentration change [31]. We used QCM in this experiment to monitor the presence of gas flow and quantify the concentration of the target compound released by this system for each experiment. Three QCMs coated with three RTILs using the dip-coating method described in Figure 5 (dip coater: VLAST45-06-0100, THK Co., Ltd., Tokyo, Japan) were applied, and the detailed coatings information is available in Table 2. Furthermore, the commercial vector network analyzer (DG8SAQ VNWA, SDR-Kits) was used to record the frequency generated by QCM. The resonant frequency was measured in real-time based on optimizing QCM equivalent circuit parameters [47]. Afterward, the gas flows into the amperometric gas sensor array, i.e., the modified IDAs with three different RTILs; the potentiostat recorded the sensor response in a current (using EVAL-AD5940ELCZ, Analog Devices, Middlesex County, MA, USA) with limited operating voltages from −1 V to +1 V. The volumes of QCMs and IDAs chambers were 18 cm^3^ and 5.25 cm^3,^ respectively. The VNWAs, potentiostats, and FPGA boards were connected to the computer.

## 3. Results

### 3.1. Measurement System Performances

To verify analyte flowing real-time, 3 QCMs coated with 3 RTILs (i.e., [EMIM][Ac], [EMIM][Otf], and [EMIM][Cl]) were applied; the same RTILs used in IDA were also used as coatings for QCMs to provide a broad understanding regarding the sensing behavior with different transducers (for more detail, see Section 4, Discussion); however, in this section, we focused on results related to modified IDA electrode to enhance the EC sensing, hence QCMs information was only used to quantify the analyte’s concentration delivered by the odor delivery system (ODS) during measurement. Three butanol isomers were used in this experiment, i.e., 1-butanol, isobutanol, and 2-butanol. Five relative concentrations (RCs) were investigated, i.e., 0%, 25%, 50%, 75%, and 100%; RC was the relative concentration to the full scale produced by ODS with 6 mL of analyte in the vial. Another sensor based on photoionization, portable detector RAE 3000 PID, was also used to quantify the amount of concentration released by ODS; the amount of concentration for every analyte read by RAE 3000 PID (in ppm) which was equivalent to RC level from ODS is available in Figure 6a–c. As shown in Figure 6a–c, all RTILs coated on QCMs exhibited the same trends and real-time response, i.e., the higher concentration of analytes, the higher the frequency change. QCM follows Sauerbrey’s formula, where the loaded mass on its surface is directly proportional to the frequency change, and the different magnitudes of frequency were generated due to varying adsorptions of each RTIL as sensing film [31,50]. Furthermore, Figure 6d shows the representative result of the concentration dependency plot from 1-butanol using QCMs coated with three RTILs. Afterward, a linear regression was used to estimate those data points. According to Figure 6d, all the linear regression models from three different RTILs had a high coefficient of determination (R^2^), indicating their strong linear trend.

Table 3 summarizes the reproducibility of the measurement system to generate butanol isomers; five measurements per analyte at five different RC were investigated, i.e., 0%, 25%, 50%, 75%, and 100%. The reproducibility was verified numerically using mean and standard deviation obtained from sensor response obtained from 3 QCMs with different RTILs. Like in Figure 6d, the linear regression model was also used to estimate the data points; according to Table 3, all sensing films coated on QCMs, i.e., [EMIM][Ac], [EMIM][Otf], and [EMIM][Cl], had high R^2^ score (above 0.95) that confirmed their strongly linear correlation to the concentration change for 1-butanol, isobutanol, and 2-butanol. 

### 3.2. Sensor Response from Modified IDA Electrode

#### 3.2.1. Comparison of Pt/PANI/Au_2_ and Pt/PANI/Au_0_

As reported in our previous study and by other researchers, using Ag/AgCl as a reference electrode (RE) is preferable to maintain a stable reaction [36,51]. Since the RE at the commercial IDA electrode was made of platinum (Pt), we used Ag/AgCl ink as RE purchased from BAS Co., Ltd., Tokyo, Japan. The Ag/AgCl ink can be easily prepared by painting and drying on the Pt electrode, as shown in Figure 7a; the drying process took two days to get strong adhesion on the Pt electrode. In this study, we used RE made of a dried Ag/AgCl ink for Pt/PANI/Au_0_ and Pt/PANI/Au_2_. Afterward, the electrolyte using 5 µL of RTIL was dropped on the modified IDA electrode’s surface, where all the WE, CE, and RE were covered, as shown in Figure 7b.

In this study, several modified IDA electrodes were prepared with two different types, i.e., Pt/PANI/Au_2_ (with Au_2_ clusters) and Pt/PANI/Au_0_ (without Au_2_ clusters); both types applied a dried Ag/AgCl ink as RE and 5 µL of RTIL as electrolyte. These sensors follow Faraday’s Law, i.e., the generated current is proportional to the amount of analyte participating in the electrochemical reaction [6]. Three butanol isomers were used as analytes, i.e., 1-butanol, isobutanol, and 2-butanol. Three RTILs having the same cation [EMIM]^+^ and different anions [Ac]^−^, [Otf]^−^, and [Cl]^−^ were used in this experiment. Figure 8 shows the comparison results of Pt/PANI/Au_2_ and Pt/PANI/Au_0_ from five repeated measurements using 3 RTILs (i.e., [EMIM][Ac], [EMIM][Otf], and [EMIM][Cl]); the RC was 100%, the exposure time for each analyte was 5 min, and the recovery time was 10 min (N_2_ flows). According to Figure 8, Pt/PANI/Au_2_ mostly exhibited catalytic activity, meaning that the Au_2_ doping process was successfully conducted on modified IDA electrodes and also active when using RTILs as electrolytes, leading to enhanced sensitivity by possessing a higher current density compared to Pt/PANI/Au_0_. 

As shown in Figure 8, Pt/PANI/Au_2_ showed the particular catalytic activity for each RTIL in term of magnitude in the current density because the different anion chain possessed by each RTIL results in different physical and chemical property; from the three RTILs used in this experiment, [EMIM][Ac] at +0.5 V obtained the highest mean current density change, which indicates that the solubility of butanol isomers was the highest using [EMIM][Ac] compared to [EMIM][Otf] and [EMIM][Cl]. In addition, three fixed electrochemical potentials (*E*s) against RE of Ag/AgCl from each RTIL were also scanned to obtain a better sensitivity; the selection of *E*s was based on the RTIL’s EC window (Table 1) and range of operating voltages from potentiostat (EVAL-AD5940ELCZ, Analog Devices). Again, the magnitude of the current density of Pt/PANI/Au_2_ was also varied by altering the *E*s; the highest sensitivity obtained at specific *E* indicated that the amount of mass diffusion transport from the analyte taking part in the electrochemical reaction was large by applied to that specific *E*; as shown in Figure 8, the high sensitivity for [EMIM][Ac] was at +0.5 V while [EMIM][Otf] and [EMIM][Cl] were at +0.25 V. Furthermore, as shown in Figure 8, by applying a higher *E*, particularly for [EMIM][Ac] at +0.9 V, the magnitude of current density was dropped; presumably, applying a high voltage at the working electrode (moreover near its maximum operating EC window) increased the humidity due to the oxidation of water since water is a common impurity in RTILs [52,53,54], thus affecting the mass transport and decreased the sensor signal, in addition, the anion [Ac]^−^ are very hygroscopic (ability to absorb and adsorb moisture from the surrounding area) and yet so hydrophilic (i.e., strong affinity to water) compared [Otf]^−^ and [Cl]^−^ [55,56,57].

#### 3.2.2. Concentration Dependency

In this experiment, the concentration dependency was also studied to obtain the sensor characteristics; five RCs were explored, i.e., 0%, 25%, 50%, 75%, and 100% (detailed information regarding the concentration of each analyte in ppm was available in Section 3.1). Figure 9a depicts the comparison of two types of modified IDA electrodes to quantify 1-butanol gas measurements using [EMIM][Ac] at +0.5 V. According to Figure 9a, Pt/PANI/Au_2_ enhanced the sensitivity of 1-butanol gas measurements for all investigated concentrations by possessing a higher current density than Pt/PANI/Au_0_; however, even Pt/PANI/Au_2_ showed a higher current density compared to Pt/PANI/Au_0_, but it couldn’t quantify the analytes at low RC due to the detection limit. Furthermore, the concentration dependence plot from Figure 9a is shown in Figure 9b. To verify the concentration dependency trends from each RTIL, the R^2^ scores were summarized in Table 4. According to Table 4, mostly Pt/PANI/Au_2_ had a higher R^2^ score compared to Pt/PANI/Au_0_; these findings indicated that using Pt/PANI/Au_2_ could achieve more linear concentration dependency trends than Pt/PANI/Au_0_.

## 4. Discussion

In our previous research, we studied three RTILs; one was [EMIM][Ac] coated onto commercial IDA without modifying its platinum working electrode [31]. Figure 10 depicts the comparison results of average sensor response from 5 measurements; three types of electrodes were used, i.e., Pt only, Pt/PANI/Au_0_, and Pt/PANI/Au_2_, all electrodes used 5 µL of [EMIM][Ac] as electrolyte and dried Ag/AgCl ink as RE, the butanol isomers gas were 100% RC, the analyte’s exposure was 5 min, followed by its recovery time for 10 min (N_2_ flows). According to Figure 10, the modified IDA electrode of Pt/PANI/Au_2_ showed the highest current density change for all analytes at all *E*s, i.e., +0.25 V, +0.5 V, +0.9 V, which means that the catalytic properties possessed by the Au_2_ clusters contributed to the enhanced sensitivity of butanol isomers gas measurement compared to the commercial Pt IDA electrode and Pt/PANI/Au_0_.

Next, we made an array of sensors to discriminate among butanol isomers. Numerical statistic criterion, Wilk’s lambda (*Λ*), was used to quantify the discriminant capability among butanol isomers; the lower the Wilk’s *Λ*, the better the separation; detailed information regarding Wilk’s *Λ* was available here [31,58,59]. Table 5 summarizes Wilk’s *Λ* values from different sensing film combinations coated on Pt/PANI/Au_2_, generated from five repeated measurements per analyte and using full-scale of 100% relative concentration (RC). According to Table 5, the lowest Wilk’s *Λ* was obtained using different sensing film combinations, i.e., [EMIM][Ac] at +0.5 V, [EMIM][Otf] at +0.5 V, and [EMIM][Cl] at +0.5 V. Furthermore, the separation power obtained by those sensing films combinations was also tested by using two different concentrations, i.e., 100% RC and 75% RC. 

Figure 11 shows the discrimination among butanol isomers visualized using linear discriminant analysis (LDA); Figure 11a,b depicts the result using only one sensing film with different *E*s (i.e., [EMIM][Ac] at +0.25 V, +0.5 V, and +0.9 V) while Figure 11c,d depicts the result using the best obtained sensing films combination based on Table 5 (i.e., [EMIM][Ac] at +0.5 V, [EMIM][Otf] at +0.5 V, and [EMIM][Cl] at +0.5 V). For one sensing film, as shown in Figure 11a, the butanol isomers were still able to be separated; however, when two concentrations were loaded (100% RC and 75% RC), as shown in Figure 11b, the distances between the three centroid groups were closer, and the discrimination among isomers became poor. Furthermore, the results from the best obtained sensing films combination were shown in Figure 11c,d; when using only 100% RC shown in Figure 11c, the butanol isomers separation was much better compared to using one sensing film depicted in Figure 11a; in addition, when information from two different concentrations was mixed, i.e., 100% RC and 75% RC, the discrimination among butanol isomers shown in Figure 11d were still achievable, and there were no overlapped data points occurred between groups even though the centroid distances between the three group were closer; therefore, these findings indicated that sensor array could contribute to the discriminant capability among butanol isomers even different concentrations loaded. 

In this experiment, we also used another sensor type, i.e., QCMs coated with RTILs which were the same used for modified IDA electrodes; based on the results, QCM coated with [EMIM][Ac] was able to quantify 1-butanol even in small RC (Figure 6). However, in the current EC measurement shown in Figure 9, Pt/PANI/Au_2_ in [EMIM][Ac] at +0.5 V could not quantify the 25% RC of 1-butanol; this issue could be solved by applying a very thin layer of RTIL to accelerate the analyte’s diffusion transport to reach the working electrode since we know that RTIL is much more viscous than aqueous electrolytes like KOH, NaOH, etc. In addition, increasing the Au_2_ cluster density on working electrodes could also improve the sensitivity for gas sensing measurements on modified IDA electrodes.

To summarize the previous works and the novelty of this study, Table 6 provides a brief comprehensive summary of research works conducted by our research groups. Since we only used butanol isomers as target compounds in this study. Hence, the sensing performance with gas interference and gas mixtures should be explored in future work to obtain comprehensive results.

## 5. Conclusions

This study successfully demonstrated and verified the enhanced sensitivity for butanol isomers gas measurement using the Au_2_ clusters as a catalyst on IDA electrodes coated with RTILs. According to the results, Pt/PANI/Au_2_ exhibited catalytic activity by possessing a higher sensor response compared to Pt/PANI/Au_0_ using different RTILs, i.e., [EMIM][Ac], [EMIM][Otf], and [EMIM][Cl]. Each RTIL provides different physical and chemical properties, which leads to the particular catalytic activity of Pt/PANI/Au_2_. The fixed electrochemical potential against Ag/AgCl from each RTIL was also explored; the sensitivity to butanol isomers was high at +0.5 V for [EMIM][Ac] and +0.25 V for [EMIM][Otf] and [EMIM][Cl]. In addition, based on the R^2^ scores, most obtained results showed that Pt/PANI/Au_2_ had a higher R^2^ score than Pt/PANI/Au_0,_ which indicated the concentration dependency trends varying more linear using Pt/PANI/Au_2_ than Pt/PANI/Au_0_. Moreover, we found the combination of RTIL with EC potential suitable for butanol isomer discrimination. 

Since we used available commercial electrodes in this study, for further enhancement, we could fabricate a smaller electrode coated with a thin layer of RTIL to accelerate the diffusion transport of analytes to the working electrode. In addition, increasing the Au_2_ cluster density could be done by optimizing the gold deposition process. The sensor array composed of different RTILs and EC potentials successfully discriminates among butanol isomers. The combination of a number of atomic metal clusters, atomic metal type, EC potential, and type of RTIL shows a variety of sensor characteristics oriented to the olfactory sensor array [16]. We can apply the active sensing method to the developed sensors to quantify the mixture in the future [60].

## Figures and Tables

**Figure 1 sensors-23-04132-f001:**
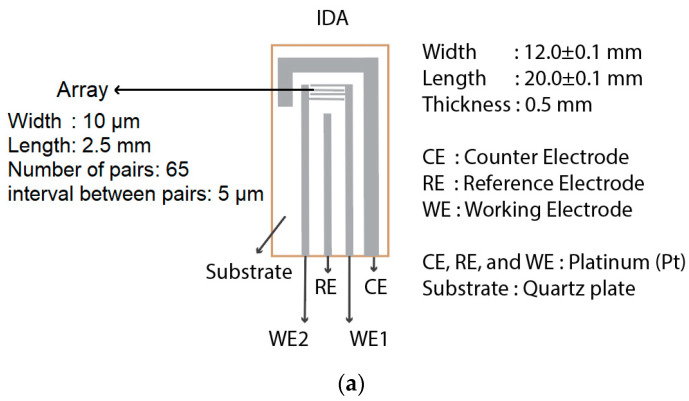
The amperometric gas sensor used in this experiment: (**a**) IDA electrode; (**b**) Modified Pt/PANI electrode; (**c**) The SEM image.

**Figure 2 sensors-23-04132-f002:**
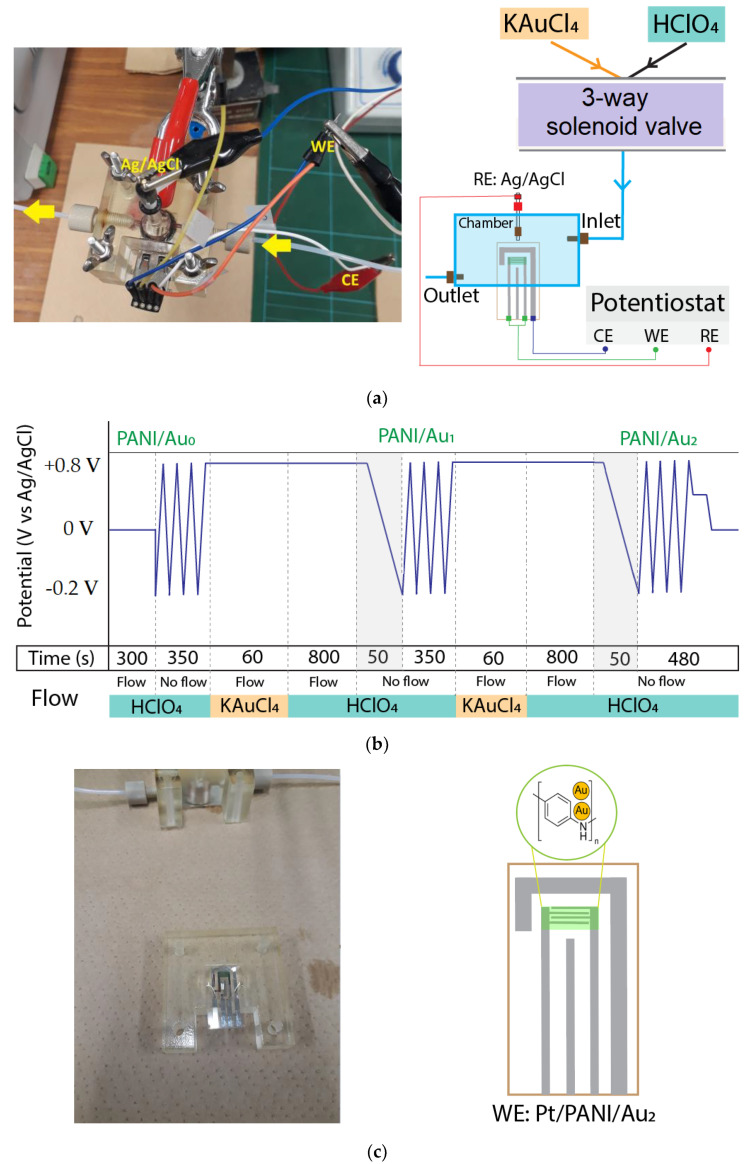
The atomic gold deposition process: (**a**) The gold deposition system; (**b**) The timing diagram; (**c**) The modified Pt/PANI/Au_2_ using IDA electrode after the gold deposition process.

**Figure 3 sensors-23-04132-f003:**
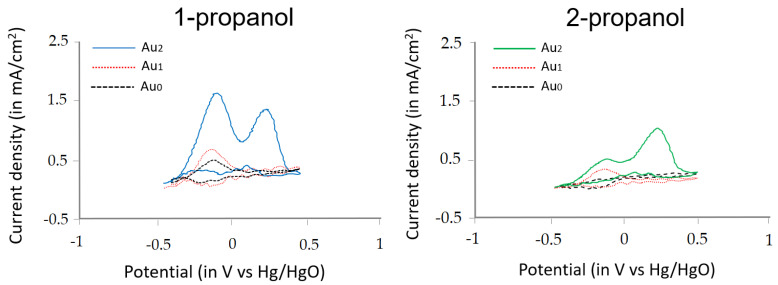
The comparison of electrooxidation from 1-propanol and 2-propanol using Pt/PANI/Au_2_, Pt/PANI/Au_1_, and Pt/PANI/Au_0_.

**Figure 4 sensors-23-04132-f004:**
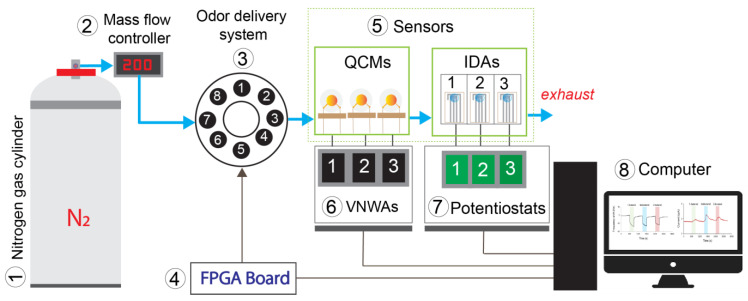
Schematic layout of the measurement system used in this experiment.

**Figure 5 sensors-23-04132-f005:**
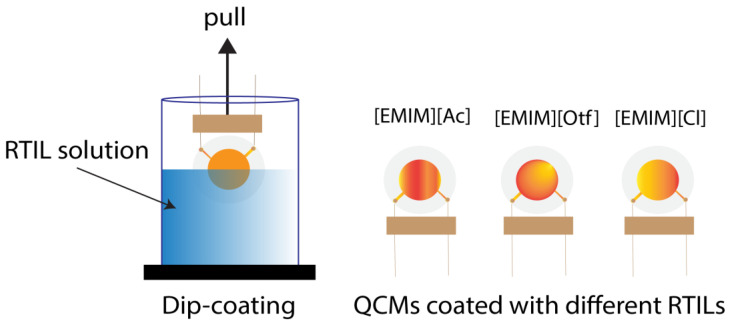
The dip-coating method for QCMs.

**Figure 6 sensors-23-04132-f006:**
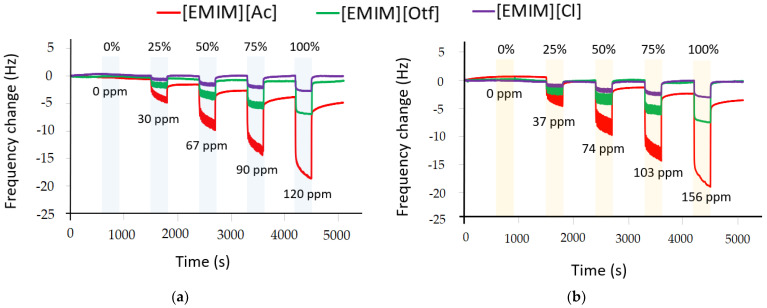
Variation of sensor responses using QCMs coated with 3 RTILs at different relative concentrations (RCs) for several analytes: (**a**) 1-butanol; (**b**) Isobutanol; (**c**) 2-butanol (each experiment has 5 min for the analyte’s exposure and 10 min for recovery (N_2_ flow)); (**d**) The concentration dependency trends for 1-butanol using QCMs coated with 3 RTILs.

**Figure 7 sensors-23-04132-f007:**
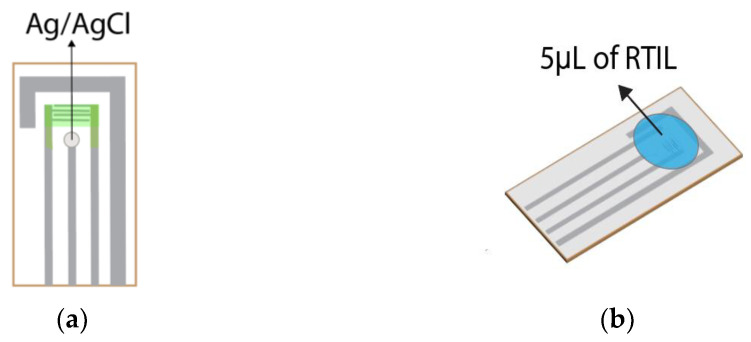
The preparation before measurement: (**a**) Applying Ag/AgCl ink as RE; (**b**) Dropping 5 µL of RTIL as electrolyte.

**Figure 8 sensors-23-04132-f008:**
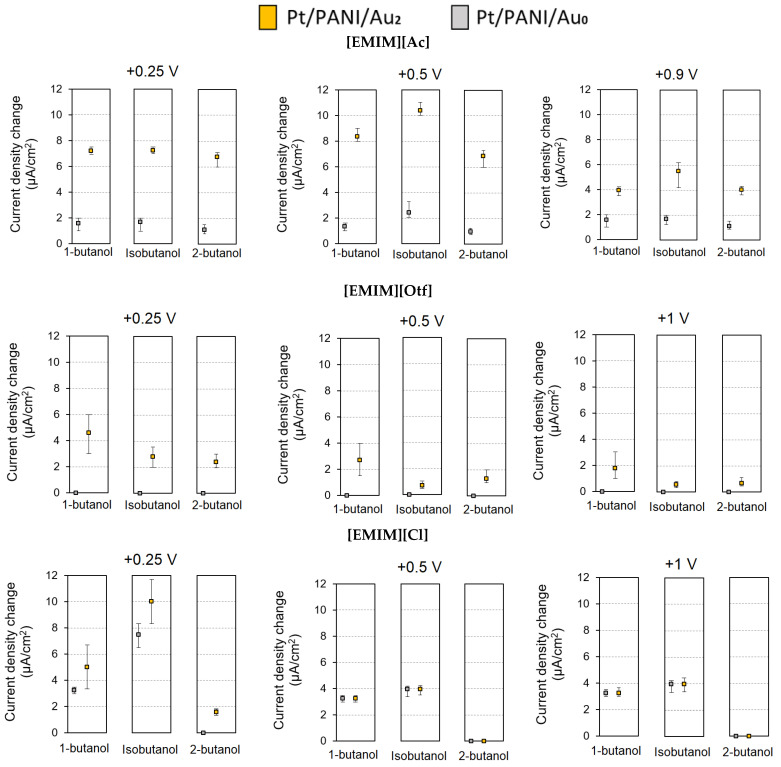
The comparison sensor response of Pt/PANI/Au_2_ and Pt/PANI/Au_0_ for butanol isomers gas measurement using three RTILs at three fixed electrochemical potentials (the rectangle is the mean value, the upper and lower line indicated the maximum and minimum value of sensor response). Furthermore, the data applying two different modified IDAs (using Pt/PANI/Au_2_ as representative) with the same RTIL was available in the Appendix A.

**Figure 9 sensors-23-04132-f009:**
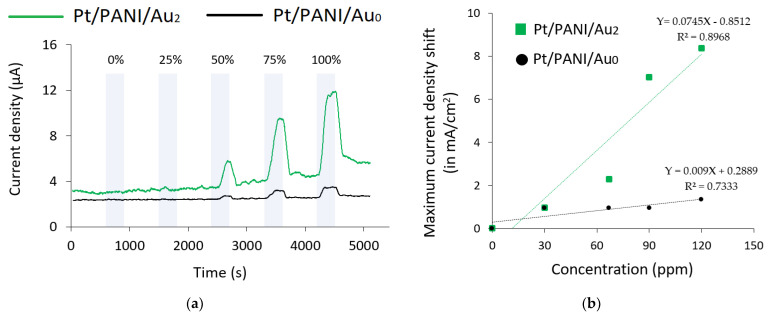
The comparison results of Pt/PANI/Au_2_ and Pt/PANI/Au_0_ 1-butanol gas measurements using [EMIM][Ac] at +0.5 V for different concentration levels: (**a**) The complete experiment; (**b**) The concentration dependence plot with the fitted linear regression model.

**Figure 10 sensors-23-04132-f010:**
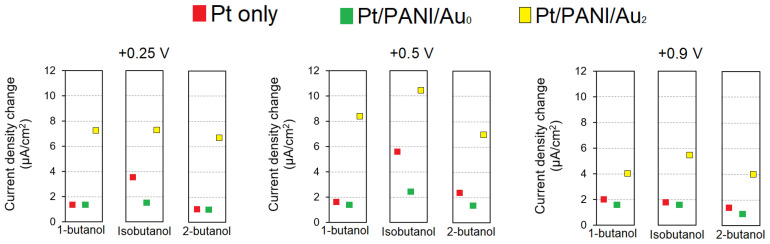
The average sensor response from five measurements at 100% RC of butanol isomers gas measurements obtained from several electrodes using [EMIM][Ac].

**Figure 11 sensors-23-04132-f011:**
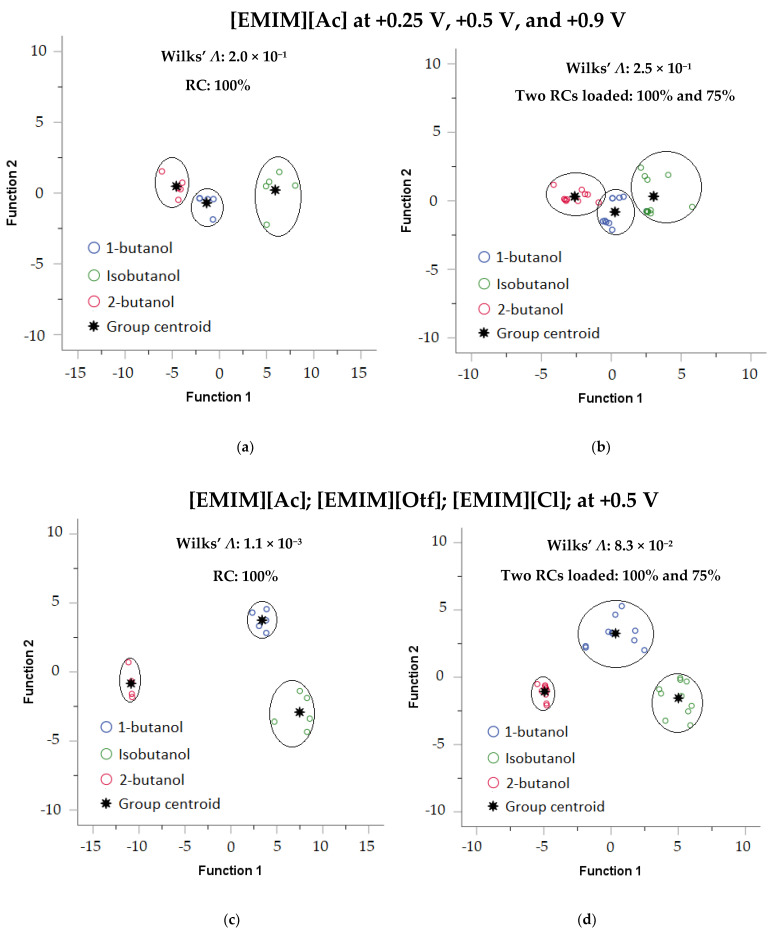
LDA was used to visualize the separation among butanol isomers from different combinations of RTIL coated on Pt/PANI/Au_2_ (**a**). Only 100% RC (one sensing film), (**b**). Two concentrations loaded, i.e., 100% RC and 75% RC (one sensing film), (**c**). Only 100% RC (three sensing films), (**d**). Two concentrations loaded 100% RC and 75% RC (three sensing films).

**Table 1 sensors-23-04132-t001:** Detailed physico-chemical properties of RTILs used in this study.

Label	CAS Number	Full Name	Viscosity, *η* (P a s)	Density, *ρ* (Kg/m^3^)	Conductivity, *ĸ* (S/m)	EC Window (V)
[EMIM][Ac]	143314-17-4	1-Ethyl-3-methylimidazolium Acetate	0.143 [37]	1099.3 [37]	0.2 [38]	−2.3 to +0.9 [39,40,41]
[EMIM][Otf]	145022-44-2	1-Ethyl-3-methylimidazolium Trifluoromethanesulfonate	0.042 [37]	1385.9 [37]	0.9 [42]	4.3 [43,44]
[EMIM][Cl]	65039-09-0	1-Ethyl-3-methylimidazolium Chloride	0.047 (a) [41]	1112 (a) [41]	0.108 [45]	-

Information of *η*, *ρ*, *ĸ* of the RTILs at 298 K (25 °C), except (a) 353.15 K (80 °C).

**Table 2 sensors-23-04132-t002:** Detailed information of sensing film coated onto QCMs.

RTIL	Solvent	Concentration (mg/mL)	Pull-Up Speed (µm/s)	ΔF (Hz)	ΔM (µg)	*d* (nm)
[EMIM][Ac]	Acetone [39]	10	1000	435	0.46	17.12
[EMIM][Otf]	Acetone [43]	9.09	1000	577	0.62	28.50
[EMIM][Cl]	Acetonitrile [48]	9.09	100	872	0.93	33.09

ΔF is the frequency change, *d* is the coating’s thickness assumed distributed uniformly on gold electrodes at QCM’s plate, and ΔM is the mass loaded after the dip-coating process calculated from Sauerbrey’s equation [49].

**Table 3 sensors-23-04132-t003:** The mean, standard deviation, and R^2^ score obtained from QCMs coated with three RTILs.

RTIL	Analyte	Mean in Hz (Standard Deviation in Hz)					R^2^
		0% RC	25% RC	50% RC	75% RC	100% RC	
[EMIM][Ac]	1-butanol	0 (0)	−4.942 (0.467)	−9.056 (0.791)	−12.7 (0.848)	−16.77 (1.532)	0.996
	Isobutanol	0 (0)	−5.08 (0.396)	−9.176 (0.548)	−13.1 (0.938)	−16.44 (1.656)	0.978
	2-butanol	0 (0)	−3.322 (0.276)	−6.102 (0.246)	−8.404 (0.620)	−10.16 (0.482)	0.971
[EMIM][Otf]	1-butanol	0 (0)	−2.138 (0.169)	−3.994 (0.148)	−5.44 (0.240)	−6.826 (0.204)	0.995
	Isobutanol	0 (0)	−2.482 (0.179)	−4.23 (0.241)	−5.782 (0.256)	−6.98 (0.258)	0.960
	2-butanol	0 (0)	−2.136 (0.276)	−3.78 (0.370)	−4.982 (0.694)	−6.22 (0.549)	0.974
[EMIM][Cl]	1-butanol	0 (0)	−0.86 (0.049)	−1.728 (0.096)	−2.146 (0.154)	−2.816 (0.093)	0.994
	Isobutanol	0 (0)	−1.04 (0.038)	−1.844 (0.104)	−2.352 (0.156)	−3.036 (0.136)	0.971
	2-butanol	0 (0)	−0.702 (0.072)	−1.31 (0.081)	−2.032 (0.167)	−2.3922 (0.183)	0.967

**Table 4 sensors-23-04132-t004:** The comparison of R^2^ score from Pt/PANI/Au_2_ and Pt/PANI/Au_0_.

RTIL	Analyte	Fixed Electrochemical Potentials (*E*s) against Ag/AgCl Ink					
		+0.25 V		+0.5 V		+0.9 V or +1 V	
		Au_2_	Au_0_	Au_2_	Au_0_	Au_2_	Au_0_
[EMIM][Ac]	1-butanol	0.947	0.52	0.896	0.73	0.964	0.78
	Isobutanol	0.89	0.628	0.857	0.846	0.89	0.825
	2-butanol	0.574	0.633	0.799	0.51	0.871	0.621
[EMIM][Otf]	1-butanol	0.968	0.5	0.919	0.5	0.685	0.5
	Isobutanol	0.799	0.5	0.52	0.5	0.74	0.5
	2-butanol	0.857	0.5	0.73	0.5	0.844	0.5
[EMIM][Cl]	1-butanol	0.8491	0.7	0.94	0.94	0.94	0.94
	Isobutanol	0.88	0.9	0.868	0.8	0.86	0.8
	2-butanol	0.5	0.5	0.5	0.5	0.5	0.5

In this table, Pt/PANI/Au_2_ and Pt/PANI/Au_0_ were labeled as Au_2_ and Au_0_, respectively. In addition, [EMIM][Otf] and [EMIM][Cl] used +1 V and [EMIM][Ac] used +0.9 V. The raw data for Table 4 are available in Appendix A.

**Table 5 sensors-23-04132-t005:** Wilk’s lambda values generated from different sensing film combinations.

Combination of Sensing Films Applied on Pt/PANI/Au_2_									Wilks’ *Λ*
[EMIM][Ac]			[EMIM][Otf]			[EMIM][Cl]			
+0.25 V	+0.5 V	+0.9 V	+0.25 V	+0.5 V	+1 V	+0.25 V	+0.5 V	+1 V	
●	●	●	×	×	×	×	×	×	2.0 × 10^−1^
×	×	×	●	●	●	×	×	×	1.28 × 10^−1^
×	×	×	×	×	×	●	●	●	4.8 × 10^−3^
●	×	×	●	×	×	●	×	×	4.9 × 10^−2^
●	×	×	●	×	×	×	●	×	5.8 × 10^−3^
●	×	×	●	×	×	×	×	●	5.8 × 10^−3^
●	×	×	×	●	×	●	×	×	3.3 × 10^−2^
●	×	×	×	●	×	×	●	×	4.2 × 10^−3^
●	×	×	×	●	×	×	×	●	4.26 × 10^−3^
●	×	×	×	×	●	●	×	×	3.85 × 10^−2^
●	×	×	×	×	●	×	●	×	4.8 × 10^−3^
●	×	×	×	×	●	×	×	●	4.8 × 10^−3^
×	●	×	●	×	×	●	×	×	2.5 × 10^−2^
×	●	×	●	×	×	×	●	×	2.7 × 10^−3^
×	●	×	●	×	×	×	×	●	2.75 × 10^−3^
×	●	×	×	●	×	●	×	×	1.4 × 10^−2^
×	●	×	×	●	×	×	●	×	1.1 × 10^−3^
×	●	×	×	●	×	×	×	●	2.1 × 10^−3^
×	●	×	×	×	●	●	×	×	2.3 × 10^−2^
×	●	×	×	×	●	×	●	×	2.9 × 10^−3^
×	●	×	×	×	●	×	×	●	2.9 × 10^−3^
×	×	●	●	×	×	●	×	×	3.5 × 10^−2^
×	×	●	●	×	×	×	●	×	1.78 × 10^−3^
×	×	●	●	×	×	×	×	●	1.78 × 10^−3^
×	×	●	×	●	×	●	×	×	3.6 × 10^−2^
×	×	●	×	●	×	×	●	×	2.10 × 10^−3^
×	×	●	×	●	×	×	×	●	2.10 × 10^−3^
×	×	●	×	×	●	●	×	×	3.7 × 10^−2^
×	×	●	×	×	●	×	●	×	2.5 × 10^−3^
×	×	●	×	×	●	×	×	●	2.5 × 10^−3^

In this table, labeled applying and not applying with ● and ×, respectively.

**Table 6 sensors-23-04132-t006:** The comprehensive summary of research studies conducted by our research groups.

Year [Ref]	Brief Summary of Research Works
2020 [32]	▪ building the atomic gold deposition system in the bulky EC sensor; ▪ verifying the odd-even pattern behavior for modified working electrodes Pt/PANI/Au_n_ (where n = 1 to 4) in aqueous electrolyte (KOH); ▪ target compounds were limited to propanol isomers only.
2020 [34]	▪ gas measurements using a bulky EC system with modified working electrodes Pt/PANI/Au_2_ and aqueous electrolyte (KOH); ▪ target compounds were ketones, alcohols, esters, and carboxylic acids.
2021 [36] 2023 [35]	▪ fabricating modified Pt/PANI/Au_2_ on a tiny EC sensor (IDA electrode); ▪ proposing the pseudo-reference electrode using a dried Ag/AgCl ink; ▪ electrolyte was only [EMIM][Otf]; ▪ fixed EC potential applied to obtain a rapid and real-time gas measurement (one fixed EC potential explored); ▪ target compounds were limited to gaseous propanol isomers only (C_3_H_8_O).
This study	▪ fabricating theAu_2_ clusters decorated Pt/PANI/ using several IDA electrodes; ▪ 3 RTILs were investigated, i.e., [EMIM][Ac], [EMIM][Otf], and [EMIM][Cl]; ▪ 3 fixed EC potentials were explored; ▪ a higher carbon chain used, i.e., gaseous butanol isomers (C_4_H_10_O); ▪ finding the possible combination of sensing films oriented to the type of RTILs and the fixed EC potentials to discriminate among the butanol isomers, i.e., 1-butanol, isobutanol, and 2-butanol.

## Data Availability

The data presented in this study are available on request.

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
