# Peer review of "Array of Miniaturized Amperometric Gas Sensors Using Atomic Gold Decorated Pt/PANI Electrodes in Room Temperature Ionic Liquid Films"

_sensors, 2023, doi:10.3390/s23084132_

Round 1
Reviewer 1 Report
Please, refer to the attached PDF document

Author Response
The authors would like to express gratitude to the reviewer for conducting the review process for our manuscript. Please find the attachment containing several files, i.e., our response, the edited manuscript oriented to the reviewer's comments, and the supplementary file.

Reviewer 2 Report
General Comments:
The manuscript is well structured and well written. The contextualization is clear and good. The research work is well structured and well defined. Authors should highlight the novelty of the proposal. Is it the miniaturization of the EC cell or is it the use of Au-PANI to distinguish butanol isomers?
Specific Comments:
Some keywords are very generic. I suggest replacing "atomic gold", "isomer", and "polyaniline" to "Miniaturized Amperometric Sensors", "Miniaturized Electrochemical Sensors", "butanol isomer detection", and "decorated PANI electrodes".
In the section "2.4. Measurement System", please indicate the test chambers volumes (QCMs chamber and IDAs chamber). These volumes have a key influence on the response time of the sensors.
Lines 224 to 233: The authors describe how the data in Table 3 are determined by mentioning "Y-axis" and "X-axis" of graphs not shown. How about replacing Table 3 with these graphs, indicating the slopes (the sensor sensitivities) and R^2 next each fitted linear regression line.
Visual information is better for the reader than lots of numbers.
I suggest the same for Table 4.
Table 3 shows values such as 0.702 Hz or 2.3922 Hz. That makes sense? What is the frequency resolution of the DG8SAQ vector network analyzer?
The LDA plots (Fig. 11) are interesting results that demonstrate the discrimination among the butanol isomers.
Author Response
The authors would like to express gratitude to the reviewer for conducting the review process for our manuscript. Please find the attachment containing several files, i.e., our response, the edited manuscript oriented to reviewer's comments, and the supplementary file.

Reviewer 3 Report
The authors have presented an amperometric gas sensing using an interdigitated array (IDA) structure. Overall, the study is interesting and supported by solid data. However, for the purpose of review, several improvements are necessary to achieve better readability of the manuscript.
1. In Fig. 3, it is not clear whether the data resulted from identical electrodes or not. There is no clear statement regarding whether the three IDAs are different, and therefore the reviewer assumes that they are all identical. Moreover, the main message of Fig. 3 is not clear. If the data are from identical electrodes, the authors should clarify whether they are trying to show the "stability" performance of three identical electrodes. If yes, it is better to compare the three curves from the three electrodes in a single chart. Unless the authors want to point out a different aspect, the Pt/PANI/Au0 and Pt/PANI/Au2 curve can be separated if the curves are too crowded.
2. It is necessary to discuss the sensing performance with gas interference or gas mixture to anticipate these scenarios.
3. The novelty of the study, which is the main contribution to the field, should be explicitly stated in the introduction. Additionally, the authors implicitly mention the objective of the study to enhance sensitivity. Therefore, a comprehensive table should be added in the discussion section that includes the results and previous related studies to emphasize the "novelty" presented in the article.
The reviewer recommends a major revision of the article.
Author Response

(The authors gave the same response as above.)

Round 2
Reviewer 1 Report
I thank the authors for taking into acount my comments and revising the Ms. accordingly.
Reviewer 3 Report
The authors addressed all the reviewer's suggestions. The manuscript has been improved. The reviewer recommends the manuscript for publication in the sensors.